# XLDA: Cross-Lingual Data Augmentation for Natural Language Inference and Question Answering

## Abstract

While natural language processing systems often focus on a single language, multilingual transfer learning has the potential to improve performance, especially for low-resource languages. We introduce XLDA, cross-lingual data augmentation, a method that replaces a segment of the input text with its translation in another language. XLDA enhances performance of all 14 tested languages of the cross-lingual natural language inference (XNLI) benchmark. With improvements of up to $4.8\%$, training with XLDA achieves state-of-the-art performance for Greek, Turkish, and Urdu. XLDA is in contrast to, and performs markedly better than, a more naive approach that aggregates examples in various languages in a way that each example is solely in one language. On the SQuAD question answering task, we see that XLDA provides a $1.0\%$ performance increase on the English evaluation set. Comprehensive experiments suggest that most languages are effective as cross-lingual augmentors, that XLDA is robust to a wide range of translation quality, and that XLDA is even more effective for randomly initialized models than for pretrained models.

## 1 Introduction

Recent work on pretraining natural language processing systems (Devlin et al., 2018; Radford et al., 2018; Howard & Ruder, 2018; Peters et al., 2018; McCann et al., 2017) has led to improvements across a wide variety of natural language tasks (Wang et al., 2018; Rajpurkar et al., 2016; Socher et al., 2013; Conneau et al., 2018). For several of these tasks, data can be plentiful for high-resource languages like English, Chinese, German, Spanish and French, but both the collection and proliferation of data is limited for low-resource languages like Urdu. Even when a large language model is pretrained on large amounts of multilingual data (Devlin et al., 2018; Lample & Conneau, 2019), languages like English can contain orders of magnitude more data in common sources for pretraining like Wikipedia.

One of the most common ways to leverage multilingual data is to use transfer learning. Word embeddings such as Word2Vec (Mikolov et al., 2013b) or GloVe (Pennington et al., 2014) use large amounts of unsupervised data to create task-agnostic word embeddings which have been shown to greatly improve downstream task performance. Multilingual variants of such embeddings (Bojanowski et al., 2017) have also shown to be useful at improving performance on common tasks across several languages. More recently, *contextualized* embeddings such as CoVe, ElMo, ULMFit and GPT (McCann et al., 2017; Peters et al., 2018; Howard & Ruder, 2018; Radford et al., 2018) have been shown to significantly improve upon aforementioned static embeddings.

BERT (Devlin et al., 2018) employs a similar strategy by using a masked version of the language modeling objective. Unlike other approaches, BERT also provides a *multilingual* contextual representation which is enabled by its shared sub-word vocabulary and multilingual training data. Often, for languages for which large amounts of data is not available, aforementioned techniques for creating embeddings (static or contextualized) is not possible and additional strategies need to be employed.

Figure 1: a) Comparing the standard monolingual approach, a naive multilingual approach that aggregates examples in various languages in a way that each example is solely in one language, and cross-lingual data augmentation (XLDA). Prediction is always in a single language. b) Two examples of XLDA inputs using the XNLI dataset.

We demonstrate the effectiveness of cross-lingual data augmentation (XLDA) as a simple technique that improves generalization across multiple languages and tasks. XLDA can be used with both pretrained and randomly initialized models without needing to explicitly further align the embeddings. To apply XLDA to any natural language input, we simply take a portion of that input and replace it with its translation in another language. This makes XLDA compatible with recent methods for pretraining (Lample & Conneau, 2019; Devlin et al., 2018). Additionally, the approach seamlessly scales for many languages and improves performance on all high- and low-resource languages tested including English, French, Spanish, German, Greek, Bulgarian, Russian, Turkish, Arabic, Vietnamese, Chinese, Hindi, Swahili and Urdu.

This paper makes the following contributions:

- We propose cross-lingual data augmenation (XLDA), a new technique for improving the performance of NLP systems that simply replaces part of the natural language input with its translation in another language.

- We present experiments that show how XLDA can be used to improve performance for every language in XNLI, and in three cases XLDA leads to state-of-the-art performance.

- We demonstrate the ability of our method to improve exact-match and F1 on the SQuAD question-answering dataset as well.

## 2 BACKGROUND AND RELATED WORK

**Multilingual Methods.** Much prior work that seeks to leverage multilingual data attempts to first train word embeddings from monolingual corpora (Klementiev et al., 2012; Zou et al., 2013; Hermann & Blunsom, 2014) and then align those embeddings using dictionaries between languages (Mikolov et al., 2013a; Faruqui & Dyer, 2014). Some instead train multilingual word embeddings jointly from parallel corpora (Gouws et al., 2014; Luong et al., 2015). Johnson et al. (2017) demonstrate how training multilingual translation in a single model can be used for zero-shot translation (i.e., for translation pairs with no parallel training data). This approach also attained state-of-the-art results for many languages. More recently, similar techniques have been adapted for extremely low resource languages (Gu et al., 2018). Neubig & Hu (2018) showed how to further fine-tune a multilingual model by explicitly using a high-resource language with a linguistically related low-resource language to improve translation quality. More recently, Conneau et al. (2017) and Artetxe et al. (2018) show how to obtain cross-lingual word embeddings through entirely unsupervised methods that do not use any dictionaries or parallel data.

**Natural Language Inference.**

The Multi-Genre Natural Language Inference (MultiNLI) corpus (Williams et al., 2017) uses data from ten distinct genres of English language for the the task of natural language inference (prediction of whether the relationship between two sentences represents entailment, contradiction, or neither). XNLI (Conneau et al., 2018) is an evaluation set grounded in MultiNLI for cross-lingual understanding (XLU) in 15 different languages that include low-resource languages such as Swahili and Urdu. XNLI

serves as the primary testbed for our proposed method, XLDA, which improves over the baseline model in all languages and achieves state-of-the-art performance on Greek, Turkish, and Urdu even without state-of-the-art pretraining Lample & Conneau (2019) introduced concurrently to our work.

**Question Answering.** We also include experiments on the Stanford Question Answering Dataset (SQuAD) (Rajpurkar et al., 2016). This dataset consists of context-question-answer triplets such that the answer is completely contained, as a span, in the context provided. For this task we translate only the training set into 4 languages using a neural machine translation system. Due to the fact that (machine or human) translation may not necessarily retain span positions, the translation of this dataset is more nuanced than the classification datasets discussed previously. For this reason, we do not tamper with the original SQuAD validation data; the test set is not publicly available either, so we constrain ourselves to a setting in which XLDA is used at training time but the target language remains English during validation and testing.

**Unsupervised Language Models.** Recently, large, pretrained, unsupervised language models have been used for XNLI, SQuAD, and the GLUE benchmark (Wang et al., 2018) to improve performance across the board. BERT (Devlin et al., 2018) pretrains deep bidirectional representations by jointly conditioning on both left and right contexts in all layers. BERT can then be fine-tuned for a specific task with an additional output layer. BERT achieved significant improvements on both XNLI and SQuAD, and is this model that serves as the base for the application of XLDA across these tasks.

**Back-translation.** Akin to XLDA, back-translation is often used to provide additional data through the use of neural machine translation systems. As a recent example, QANet (Yu et al., 2018) employed this technique to achieve then state-of-the-art results on question answering, and machine translation systems (Sennrich et al., 2016) regularly use such techniques to iteratively improve through a back-and-forth process. XLDA also translates training data from a source language into a target language, but XLDA does not translate back into the source language. In fact, experimental results show that XLDA provides improved performance over using even the original source data, let alone a noisier version provided through back-translation. This indicates that signal in multiple languages can be beneficial to training per se rather than only as an intermediary for back-translation.

## 3 XLDA: CROSS-LINGUAL DATA AUGMENTATION

Let $\mathcal{D} = \{(x_i, y_i, z_i)\}$ be a dataset of input text sequences $x_i$ and $y_i$ with labels $z_i$. We create a new dataset $\mathcal{D}_{lm} = \{(x_i^{(l)}, y_i^{(m)}, z_i)\}$, where $x_i^{(l)}$ is the translation of $x_i$ into language $l$ and $y_i^{(m)}$ the translation of $y_i$ into language $m$ by a neural machine translation system.

When $l = m$, all inputs are in the same language, which is the monolingual setting. When a training set is the union of multiple $\mathcal{D}_{ll}$, this is the disjoint, multilingual setting (DMT) as multiple languages are used for training, but each individual example only contains one language. When $l \neq m$, each input is in a different language, which is the cross-lingual setting. Experiments below demonstrate that the cross-lingual setting can improve upon the monolingual and DMT settings for natural language inference and a combination of monolingual and cross-lingual training yields the best results for span-extractive question answering.

## 4 EXPERIMENTS AND RESULTS

We experiment with using various subsets of $\mathcal{D}_{XLDA}$ and demonstrate empirically that some provide better learning environments than the monolingual and disjoint, multilingual settings. First, we provide details on how different translation systems $\mathcal{T}$ were used to generate cross-lingual datasets for both tasks under consideration.

### 4.1 CROSS-LINGUAL DATA

The first task we consider in these experiments is MultiNLI (Williams et al., 2017). The XNLI (Conneau et al., 2018) dataset provides the disjoint, multilingual version of MultiNLI necessary for XLDA. To create the XNLI dataset, the authors used 15 different neural machine translation systems (each for a different language pair) to create 15 separate single language training sets. The validation and test sets were translated by humans. The fact that XNLI is aligned across languages for each

sentence pair allows our method to be trained between the examples in the 15 different training sets. Since XLDA only pertains to training, the validation and test settings remain the same for each of the 15 languages, but future work will explore how this method can be used at test time as well. Because we have human translated validation and test sets for XNLI, it is the primary task under examination in our experiments.

We follow a similar process for the Stanford Question Answering Dataset, SQuAD (Rajpurkar et al., 2016). Given SQuAD is a span-extraction problem, translation of the training set required special care. Questions, context paragraphs, and gold answers were translated separately. We then used exact string matching between the translated answers and the translated context paragraphs to determine if the translated answer could still be found in the translated context. If the answer exists, we take its first instance as the ground truth translated span. 65% of German, 69% of French, 70% of Spanish, and 45% of Russian answer spans were recoverable in this way. To translate the rest of the questions we placed a special symbol on both sides of the ground truth span in the English context paragraph before translation. Occasionally, even with this special symbol, the translated answers could not be recovered from the marked, translated context. Additionally, the presence of the symbol did influence gender and tense in the translation as well. Using this approach on examples that failed span-recovery in the first phase, we were able to recover 81% of German, 96% of French, 97% of Spanish, and 96% of Russian answer spans. However, adding this additional data from the marked, translations led to worse performance across all of the languages. Hence, we only use the translated examples from the first phase for all training settings below, effectively reducing training set size. The validation and test splits for SQuAD remain in English alone[1]. This still allows us to explore how XLDA can be used to leverage multilingual supervision at train time for a single target language at validation time.

## 4.2 MODELS

For XNLI, we demonstrate that XLDA improves over the multilingual BERT model ($BERT_{ML}$) fine-tuned for different languages using a trained classification layer (Devlin et al., 2018). In XNLI, there are two inputs, a premise and a hypothesis. The BERT model takes as input both sequences concatenated together. It then makes a prediction off of a special CLS token appended to the start of the combined input sequence. We experiment with a variety of settings, but the primary evaluation of XLDA shows that replacing either one of the XNLI inputs with a non-target language can always improves performance over using only the target language throughout. These experiments are discussed in detail in sections 4.3-4.5. Similar experiments with an LSTM baseline that is not pretrained are outlined in 4.6, which demonstrates that XLDA is also effective for models that are not pretrained or as complex as BERT.

For SQuAD, we demonstrate that XLDA also improves over $BERT_{ML}$ fine-tuned by using only two additional parameter vectors: One for identifying the start token and for the end token again following the recommendations in Devlin et al. (2018). Experiments with these BERT models demonstrate that XLDA can improve even the strongest multilingual systems pretrained on large amounts of data.

For all of our experiments we use the hyperparameters and optimization strategies recommended by Devlin et al. (2018). For both datasets the learning rate is warmed up for $10\%$ of the total number of updates (which are a function of the user specified batch size and number of epochs) and then linearly decayed to zero over training. For XNLI the batch size is 32, learning rate is 2e-5 and number of epochs is 3.0. For SQuAD the batch size is 12, learning rate is 3e-5 and number of epochs is 3.0.

## 4.3 PAIRWISE EVALUATION

Our first set of experiments comprise a comprehensive pairwise evaluation of XLDA on the XNLI dataset. The results are presented in Figure 2a. The language along a row of the table corresponds to the language evaluated upon. The language along a column of the table corresponds to the auxiliary language used as an augmentor for XLDA. Diagonal entries are therefore the validation scores for the standard, monolingual training approach. Numbers on the diagonal are absolute performance. Numbers on the off-diagonal indicate change over the diagonal entry in the same row. Through color normalization, deep green represents a large relative improvement of the number depicted over the standard approach (the diagonal entry in the same row). Deep red represents a larger relative decrease

---

[1]The test set is not publicly available, so it could not be translated as XNLI was.

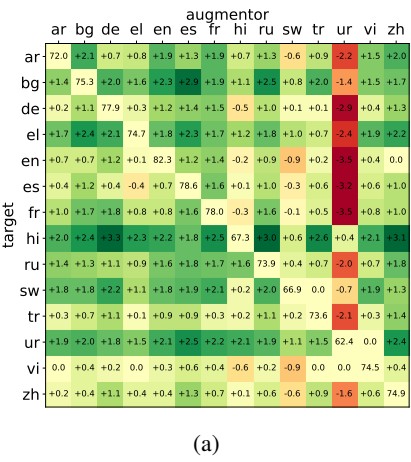
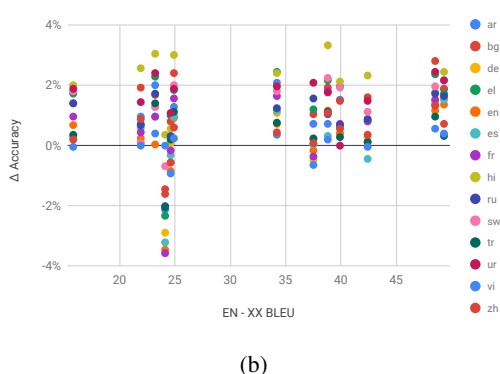

(a)                    (b)

Figure 2: a) A comprehensive pairwise evaluation demonstrates that all languages have a cross-lingual augmentor that improves over monolingual training. Rows correspond to the evaluated language. Columns are cross-lingual augmentors. Numbers on the diagonal are performance of monolingual training. Numbers on the off-diagonal indicate change over the diagonal entry in the same row. Red to green indicates stronger improvement over the baseline with XLDA. b) The BLEU score of the translation system has little effect on a language's performance as a cross-lingual augmentor. The y-axis shows the change XNLI validation accuracy, and the x-axis shows the BLEU scores of the NMT systems corresponding to the language used as a cross-lingual augmentor. The translation qualities, in BLEU, for the English to language X systems were as follows ar:15.8, bg:34.2, de:38.8, el:42.4, es:48.5, fr:49.3, hi:37.5, ru:24.9, sw:24.6, tr:21.9, ur:24.1, vi:39.9, zh:23.2.

compared to the standard approach. Mild yellow reflects little to no improvement over the standard approach.

**There exists a cross-lingual augmentor that improves over the monolingual approach.** First note that the highest performance for each row is off-diagonal. This demonstrates the surprising result that for every target language, it is actually better to train with cross-lingual examples than it is to train with monolingual examples. For example, if Hindi is the target language, the standard monolingual approach would train with both premise and hypothesis in Hindi, which gives a validation performance of 67.3%. XLDA with German in this case includes only examples that have either the premise or hypothesis in German and the other in Hindi. Therefore, there are no examples in which both premise and hypothesis are in the same language. This improves performance by 3.3% to 70.6%. Similarly, for every language a XLDA approach exists that improves over the standard approach. Hindi augmented by German represents the strongest improvement, whereas Vietnamese augmented by Spanish represents the weakest at a 0.6% improvement over the standard approach.

**Most languages are effective augmentors.** With the exception of Urdu, Swahili, Hindi, and Greek, the remaining 10 languages provided a nonnegative performance improvement as a augmentor for each of the 14 target languages. This demonstrates that as long as one avoids the lowest resource languages as augmentors, it is likely that XLDA will improve over the standard approach. It also demonstrates that with limited translation resources, one should translate into relatively higher resource languages for training machine translation systems (e.g. Spanish, German, French, English).

**Lower resource languages are less effective augmentors, but benefit greatly from XLDA.** Examining this further, it is clear that languages that are often considered relatively lower resource in the machine translation community tend to be less effective cross-lingual augmentors. For example, the abundance of red in the Urdu column reveals that Urdu is often the worst augmentor for any given language under consideration. On average, augmenting with Urdu actually hurts performance by 1.8%. On the other hand, looking at the row for Urdu reveals that it often benefits strongly from XLDA with other languages. Similarly, Hindi benefits the most from XLDA, and is only a mildly successful augmentor.

**XLDA is robust to translation quality.** In creating the XNLI dataset, the authors used 15 different neural machine translation systems with varying translation qualities according to BLEU score evaluation. The translation qualities are present in the caption of Figure 2b, which shows that even when controlling for BLEU score, most languages can be used as effective augmentors.

## 4.4 A Greedy Algorithm for XLDA

Given that the pairwise evaluation of Section 4.3 reveals that most languages are effective cross-lingual augmentors, we turn to the case in which we would like to maximize the benefits of XLDA by using multiple augmenting languages. In these experiments, we use a simple greedy approach to build off of the pairwise results. For any given target language, we sort the languages in order of decreasing effectiveness as an augmentor (determined by relative improvement over the standard approach in the pairwise setting). We start with the augmentor that is most effective for that target language and add augmentors one at a time in decreasing order until we reach an augmentor that hurt. The results are presented in Figure 3 where each subplot is corresponds to performance on a target language as number of cross-lingual augmentors increases

| Lang. | $\text{BERT}_{ML}$ | Greedy XLDA | $+\Delta$ |
|-------|------|------|-----|
| ar | 71.4 | 75.0 | 3.6 |
| bg | 75.9 | 79.1 | 3.2 |
| de | 76.1 | 78.7 | 2.6 |
| el | 74.8 | **78.4** | 3.6 |
| en | 81.5 | 83.4 | 1.9 |
| es | 79.0 | 80.4 | 1.4 |
| fr | 78.3 | 78.8 | 0.5 |
| hi | 67.2 | 71.9 | 4.7 |
| ru | 74.6 | 76.7 | 2.1 |
| sw | 65.2 | 70.0 | 4.8 |
| tr | 72.2 | **75.1** | 2.9 |
| ur | 63.0 | **65.9** | 2.9 |
| vi | 72.6 | 76.1 | 3.5 |
| zh | 74.6 | 76.0 | 1.4 |

Table 1: Test results on XNLI comparing multilingual BERT without and with Greedy XLDA. At the time of our work, multilingual BERT was the current state-of-the-art, but work concurrent to ours now holds state-of-the-art on all but the bolded languages (Lample & Conneau, 2019).

**Greedy XLDA always improves over using the single best cross-lingual augmentor.**

For every target language, the greedy approach improves over the best pairwise XLDA by a minimum of 0.9% and provides a minimum of 2.1% improvement over the original standard approach that does not use any form of XLDA. Somewhat surprisingly, it is not the case that more data always helps for XLDA. Most languages have peak validation performance with fewer than the total number of augmentors that benefited in pairwise evaluation. In the best cases (Russian, Greek, and Arabic), greedy XLDA improves over the best pairwise XLDA by more than 2%. When compared to the standard approach, greedy XLDA improves by as much as 4.9% (Hindi).

## 4.5 Targeted XLDA

Since most languages are effective augmentors and few are actually harmful to performance (Section 4.3), we now consider the setting in which a comprehensive pairwise evaluation and greedy

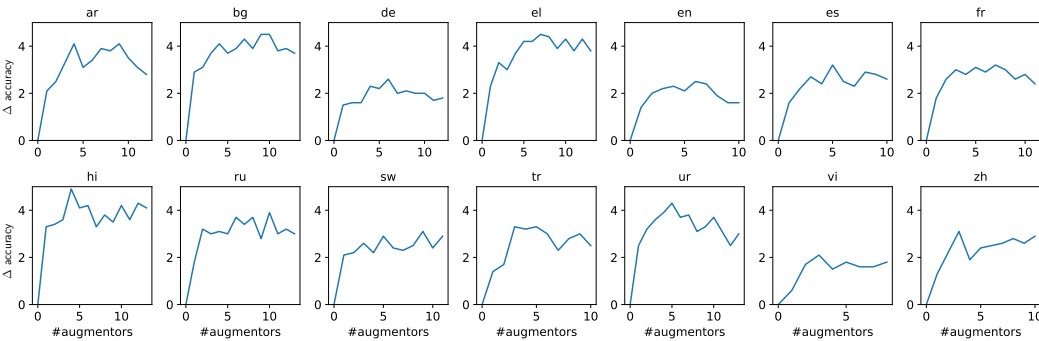

Figure 3: Greedily adding cross-lingual augmentors to $\text{BERT}_{ML}$ based on the pairwise evaluation.

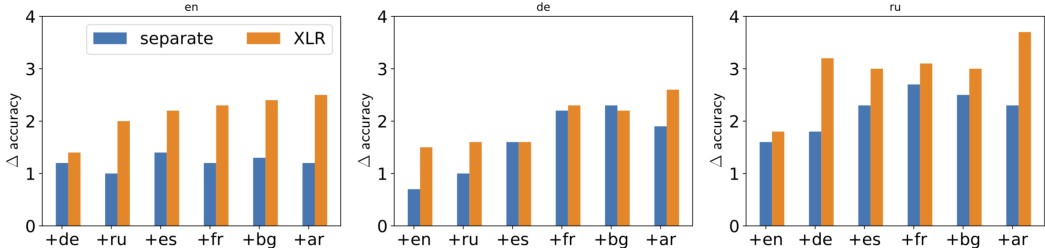

Figure 4: Both cross-lingual data augmentation (XLDA) and disjoint, multilingual training (DMT, see Section 3) improve over monolingual training, but XLDA provides greater improvements as more augmentors are used.

search cannot be done. We use this setting to evaluate whether XLDA improves over the disjoint, multilingual setting described in Section 3. Recall that in the XLDA setting each input to the model is in a different language for each example. In the disjoint, multilingual setting, each input is in the same language, but examples themselves may come from different languages.

**The 'cross' in cross-lingual is crucial.** Figure 4 shows three selected target languages (English, German, and Russian) as well as six augmentors (English, German, Russian, French, Bulgarian, and Arabic). We compare, for each target language, how incrementally adding augmentors influences performance in both the cross-lingual and the disjoint, multilingual settings. It is clear that while both improve over the monolingual setting, XLDA provides a much greater improvement as additional augmentors are used. This demonstrates that it is indeed the cross-over of languages in XLDA that makes it so effective. It is not as effective to train with translated versions of the training datasets without cross-linguality in each example.

## 4.6   XLDA WITHOUT BERT

In order to test whether the benefits of XLDA come only from the multilinguality of the BERT model, we partially replicate the pairwise evaluation of Section 4.3 for an LSTM baseline. For this baseline, we use the same tokenization as $BERT_{ML}$. We also use the embeddings from $BERT_{ML}$, but we keep them fixed. This NLI model reads the input with a two-layer BiLSTM, projects the outputs from the final layer to a lower dimension, max-pools, and passes that through a final classification layer.

**XLDA is equally effective for randomly initialized and pretrained models.** As seen in Figure 5a, the LSTM baseline sees gains from XLDA as substantial as $BERT_{ML}$. In the best case (Greek augmented by German), performance improved by 3.3%, just as high as the highest gain for $BERT_{ML}$.

**Greedy XLDA is more effective for randomly initialized models than pretrained models.** As can be seen in Figure 5b, the LSTM baseline sees gains from greedy XLDA that are even greater than they were for $BERT_{ML}$. German's XLDA performance was improved by 3.3% over using pairwise XLDA alone. This represents an absolute improvement of 5.3% over the standard, monolingual approach. In the best case (Greek), the absolute gain was 5.5% and in the worst case it was 4.0%. This demonstrates that greedy XLDA is a powerful technique that can be used with pretrained and randomly initialized models alike.

## 4.7   XLDA FOR SQUAD

We continue with experiments on the Stanford Question Answering Dataset (SQuAD). In this setting, evaluation is always performed with English contexts and questions. In Table 2, validation results are depicted that vary over which languages were used during training for either the question or the context. The en-en row represents the case in which only English inputs are used.

In the first group of four rows of Table 2, XLDA is only applied to translate the contexts. In the second group of four rows of Table 2, we also translate the question. When French is used as the cross-lingual augmentor over the context input, we see an improvement of 0.5 EM and 1 F1. We also ran these experiments with Russian, Spanish, and German, each of which proved to be effective cross-lingual augmentors over the context.

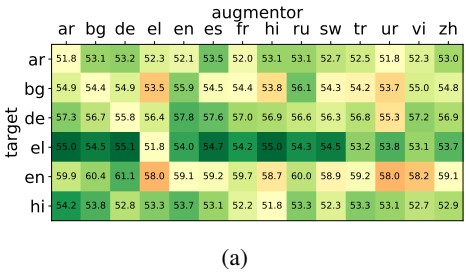
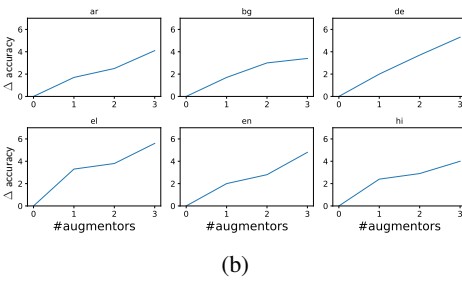

(a)                                                          (b)

Figure 5: a) XLDA is on average even more effective for a given language when used on randomly initialized models compared to large, pretrained ones. b) Greedily adding cross-lingual augmentors to an LSTM baseline.

When cross-lingual augmentors are used over the question input as well, we still often see improvement over the baseline, but the improvement is less than when using XLDA only over the context channel. This is in keeping with the findings of Asai et al. (2018), which show that the ability to correctly translate questions is crucial for question answering. In other words, SQuAD is sensitive to the translation quality of the question, and it is not surprising that machine translations of the questions are less effective than translating the context, which is less sensitive.

Error analysis on SQuAD provides insight into how XLDA improves performance. By inspection of 300 examples, we found the baseline $\text{BERT}_{ML}$ model often makes mistakes that suggest a faulty heuristic based on fuzzy, pattern matching. The model regularly depends on key words from the question in the context as well. When these keywords are nearer to plausible, incorrect spans than they are to the correct span, the model chooses the incorrect spans. In one example, the context contains the text *"...The Super Bowl 50 halftime show was headlined by the British rock group Coldplay with special guest performers Beyonce and Bruno Mars, who headlined the Super Bowl XLVII and Super Bowl XLVIII halftime shows, respectively..."*, the question is *"What halftime performer previously headlined Super Bowl XLVII?"*, and the true answer is *"Beyonce"*. The baseline model outputs *"Bruno Mars"* seemingly because it is closer to key words *"Super Bowl XLVII"*, but the XLDA model correctly outputs *"Beyonce"*. Because these models rely on attention (Bahdanau et al., 2014; Vaswani et al., 2017) between the question and context sequences, word-similarity-based matching seems to localize keywords. Such examples are often correctly answered after XLDA is used during training. This suggests that translation of the context into a different language during training (via XLDA) breaks the strong dependence on the word-similarity-based heuristic. XLDA instead forces the model to consider the semantics of the context instead.

| Ques. | Context | EM | nF1 |
|---|---|---|---|
| en | en | 81.7 | 88.5 |
| en | en,fr | +0.5 | +1.0 |
| en | en,es | +0.4 | +0.8 |
| en | en,ru | +0.2 | +0.7 |
| en | en,de | 0.0 | +0.8 |
| en,fr | en,fr | +0.1 | +0.6 |
| en,es | en,es | -0.3 | +0.3 |
| en,ru | en,ru | -0.5 | +0.1 |
| en,de | en,de | -0.4 | 0.0 |

Table 2: SQuAD validation results when questions and contexts are both in English. In the first row is the baseline: English questions and contexts are used in training. The row for English questions and both English and French contexts demonstrates XLDA improves performance by as much as 1 nF1.

## 5 CONCLUSION

We introduce XLDA, cross-lingual data augmentation, a method that improves the training of NLP systems by replacing a segment of the input text with its translation in another language. We show how reasoning across languages is crucial to the success of XLDA. We show the effectiveness of the approach with both pretrained models and randomly initialized models. We boost performance on all languages in the XNLI dataset, by up to 4.8%, and achieve state of the art results on 3 languages including the low resource language Urdu. Further investigation is needed to understand the causal and linguistic relationship between XLDA and performance on downstream tasks.

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
