# OpenReview forum: "XLDA: Cross-Lingual Data Augmentation for Natural Language Inference and Question Answering"
_ICLR.cc/2020/Conference — Reject_

### Official Review · AnonReviewer1 · 2019-10-22
**Official Blind Review #1**

**Rating:** 1

**Review:**

The paper proposes a cross-lingual data augmentation method to improve the language inference and question answering tasks. The core idea is to replace a port of the input text (such as one of the sentence in a sentence pair in the language inference tasks) with its translation in another language. The authors empirically show that deploying the XLDA data augment improves the baseline methods for both the XNLI language inference data sets and the SQuAD task.

The paper is generally easy to follow and the experiments show promising results. Nevertheless, I have a concern with the comparison baseline used in the paper, as follows.

The authors indicate that their work is close to back-translation at the end of the related work section. Also, the authors provide their hypotheses about the possible disadvantages of using back-translation as data augmentation, compared to the proposed work here. However, the authors did not provide experimental studies to support that. To me, the proposed data augmentation strategy is very similar to the back-translation, so I would expect the authors to use that as a comparison baseline in their experimental studies. Without those results, it is difficult to judge the contributions of the proposed approach.

Also, the paper could be significantly improved if the authors could provide some analysis/observations on why replacing one of the sentences in the sentence pair for the inference tasks helps? In this sense, using some other data augmentation methods as comparison baselines would be very helpful. For example, simply drop some words in the sentence pair.

I found the observations in the paper interesting, but the lack of comparison baselines makes me think the paper in its current form is not ready yet.


**Experience Assessment:**

I have published one or two papers in this area.

**Review Assessment: Checking Correctness Of Derivations And Theory:**

N/A

**Review Assessment: Checking Correctness Of Experiments:**

I carefully checked the experiments.

**Review Assessment: Thoroughness In Paper Reading:**

I read the paper thoroughly.

---

### Official Review · AnonReviewer2 · 2019-10-23
**Official Blind Review #2**

**Rating:** 8

**Review:**

Summary: This paper proposes to augment crosslingual data with heuristic swaps using aligned translations, somewhat like what bilingual humans do in code-switching. I think this paper investigates a neat extension of the XNLI dataset, which is in fact the sort of thing it was created to enable! It also looks at SQuaD translations (but, I'd have preferred a bit more depth on one of these datasets over having both, but I understand why you made this rhetorical choice).

Your augmentation extension to XNLI also uncovers a bunch of surprising results, like that code-switched utterances help models do better than monolingual ones! My main issue, if i had to find one, is that the paper doesn't try to offer (even possible) explanations for the unexpected results; maybe try to find space for more of these in a discussion section? Finally, this paper is really fun and well written, thanks for the effort! I'm going to leave a bunch of questions: it would be cool to see some in the final, but if they don't fit, you can consider them for a follow up.

Questions:
-Are all "portions" full sentences? Did performance change based on which "portions" you swapped?  In the human code-switching literature, there are syntactic generalizations about what gets switched. If you analyze the swapping, you could figure out which parts of the sentence (say, verb phrases v. prepositional phrases, beginning v. middle v. end, etc.) mattered more for NLI performance. I'd love to know the answer to that question!
-you say this: "The BLEU score of the translation system has little effect on a language’s performance as a cross-lingual augmentor. " Any ideas on why?
-you also say this: "for every language a XLDA approach exists that improves over the standard approach", what a tantalizing statement! Why did that happen?!
-Are there any generalizations over whether typologically similar languages are better augmentors for each other than they are for really different ones? I feel like if you could redo your XLR method (fig. 4) by adding augmentors in order from most similar to least (or vice versa), and you might find the answer to this.
-for XNLI, I'd love to see if you have differences by label (maybe in an appendix?)

Small Notes:
-the text in fig1 should be bigger.
-too many Ms and Ls, you had me chuckling at all the acronym puns!
-define "augmentor" somewhere

**Experience Assessment:**

I have published one or two papers in this area.

**Review Assessment: Checking Correctness Of Derivations And Theory:**

I did not assess the derivations or theory.

**Review Assessment: Checking Correctness Of Experiments:**

I assessed the sensibility of the experiments.

**Review Assessment: Thoroughness In Paper Reading:**

I made a quick assessment of this paper.

---

### Official Review · AnonReviewer3 · 2019-10-30
**Official Blind Review #3**

**Rating:** 3

**Review:**

The paper provides an analysis of a cross-lingual data augmentation technique dubbed XLDA, which consists of replacing parts of an input text with its translation in another language. Building on the mBERT approach, the authors show that at fine-tuning time it is beneficial to augment the training set of XNLI with cross-lingual hypotheses and premises instead of in-language pairs. For each language in XNLI, they show results by augmenting with each of the 14 other languages in the dataset, and show significant improvements over per-language performance.

The paper explores an interesting idea and shows that cross-lingual data augmentation works well. However, their analysis is limited to the XNLI and the Squad dataset, which do not cover a suitable range of tasks to fully conclude on the importance of XLDA for generally improving NLU tasks. It would have been interesting to show the effect of cross-lingual data augmentation for other GLUE tasks by augmenting the datasets with machine translation for instance. And also compare this model on these tasks to the monolingual back-translation approach, similar to https://arxiv.org/abs/1904.12848 . Applying XLDA with the latest open-source XLM models from the cross-lingual language model pretraining paper which obtain higher performance than the multilingual BERT would also make the results more convincing. While I share the excitement of using cross-lingual models to improve monolingual performance, I also feel like this paper lacks novelty and further evaluation to be accepted at ICLR, and would be more suited in a more NLP-focused venue.

**Experience Assessment:**

I have published in this field for several years.

**Review Assessment: Checking Correctness Of Derivations And Theory:**

I assessed the sensibility of the derivations and theory.

**Review Assessment: Checking Correctness Of Experiments:**

I carefully checked the experiments.

**Review Assessment: Thoroughness In Paper Reading:**

I read the paper thoroughly.

---

### Decision · Program_Chairs · 2019-12-19

**Decision:**

Reject

**Comment:**

The authors provide an analysis of a cross-lingual data augmentation technique which they call XLDA. This consists of replacing a segment of an input text with its translation in another language. They show that when fine-tuning, it is more beneficial to train on the cross-lingual hypotheses than on the in-language pairs, especially for low resource languages such as Greek, Turkish and Urdu. The paper explores an interesting idea however they lack comparison with other techniques such as backtranslation and XLM models, and would benefit from a wider range of tasks. I feel like this paper is more suitable for an NLP-focussed venue.